# Incidental Calcifications of Carotid and Vertebral Arteries: Frequency and Associations in Pediatric Population

**DOI:** 10.3390/diagnostics15101263

**Published:** 2025-05-15

**Authors:** Turkhun Cetin, Gokce Cinar, Berna Ucan, Fulya Memis, Baris Irgul, Sonay Aydin

**Affiliations:** 1Department of Radiology, Erzincan Binali Yidirim University, Erzincan 24100, Turkey; turkhuncetinmd@gmail.com (T.C.); sonay.aydin@erzincan.edu.tr (S.A.); 2Department of Radiology, Ankara Etlik City Hospital, Ankara 06170, Turkey; gokcecinar@gmail.com (G.C.); bernaucan@gmail.com (B.U.); 3Department of Internal Medicine, Erzincan Binali Yidirim University, Erzincan 24100, Turkey; fulyamemis@gmail.com

**Keywords:** carotid artery calcifications, vertebral artery calcifications, incidental cranial calcifications

## Abstract

**Background:** Calcifications in the carotid and vertebral arteries may be present on cranial and temporal bone CT imaging of pediatric patients. Few studies have investigated the frequency, location, and patterns of carotid artery calcifications in this age group. However, these studies are outdated and do not include data on the vertebral artery. The aim of this study was to determine the frequency, location, and pattern of incidental carotid and vertebral artery calcifications on cranial CT and temporal bone CT images in children under 15 years of age. We also aimed to investigate possible associations between these calcifications and various diseases. **Methods:** A total of 300 CT images of the cranial and temporal bone of 300 pediatric patients were retrospectively evaluated for the presence of calcification in the carotid and vertebral arteries. The evaluation included determining the presence of calcification in the artery, the pattern of calcification, the degree of calcification, and its anatomical location. **Results:** In the current study, 300 CT images were analyzed, and calcifications were found in the vertebral artery in 17 patients (5.6%) and the carotid artery in 82 patients (27.3%). The supraclinoid segment and the carotid siphon regions are the most common locations of carotid artery calcifications, with 62 patients (20.7%). The V4 segment is also the most common location for vertebral artery calcifications, with 15 patients (5%). Focal punctate calcification is the most common pattern (65 patients, 21.7%). Incidental carotid and vertebral artery calcifications did not correlate with other diseases. **Conclusions:** Carotid and vertebral artery calcifications are common incidental findings in pediatric patients. In our study, no association was found between other diseases and incidental carotid and vertebral artery calcifications.

## 1. Introduction

Incidental intracranial arterial calcifications detected on CT imaging of adult patients have attracted much attention due to their potential predictive value for atherosclerotic disease, hypercholesterolemia, diabetes mellitus, heart disease, carotid artery stenosis, and stroke [1,2,3,4,5,6].

Risk factors include diabetes mellitus, hypercholesterolemia, a history of cardiovascular disease in both sexes, excessive alcohol and smoking in males, and hypertension in women. The prevalence and volume of carotid artery calcification increases with age. Calcification is inversely related to mean flow velocity and associated with measures of vascular stiffness, such as pulse pressure, aortic pulse wave velocity, and pulsatility index [1,7,8,9]. Calcification has a significant negative predictive value for carotid bifurcation stenosis, but it is not a reliable indicator of stenosis in stroke suspects [10]. In stroke patients, there was a weak association between severe intracranial internal carotid artery stenosis and calcium burden [11]. After subarachnoid hemorrhage, severe calcification is associated with lower rates of vasospasm [10].

Age, pulse pressure, and a family history of vascular disease are all associated with carotid artery calcification, both intimal and medial. In addition, medial calcification is associated with diabetes mellitus and a history of vascular disease, while intimal calcification is associated with smoking and high blood pressure [12]. Intracerebral artery calcification can occur in any part of the brain and is commonly seen on CT imaging in the general population. In the large arteries, calcifications are a strong predictor of poor clinical outcomes and often become more common with age [1].

Just before entering the brain parenchyma, the ICA and its branches form a tortuous segment, often referred to as the carotid siphon, roughly corresponding to segments C4 and C5 (Figure 1). It is thought that this elastic and tortuous siphon compensates for the increased pulse pressure caused by the passage of the intracranial internal carotid artery through the skull [13]. In the general population, calcification of the carotid and vertebral arteries in the cervical and cranial regions is a common incidental finding on CT imaging [1]. The prevalence increases with age and is most common in the ICA [1].

In pediatric age groups, calcification in the ICA is more common in the supraclinoid segment and the carotid siphon regions [14,15]. And, calcification in the vertebral artery is more common in the V4 segment [14].

In population-based cohort studies, the prevalence of supraclinoid segment and siphon calcifications on CT has been reported to be 6% in pediatric age groups. For the vertebral artery, although no studies have been conducted in specific pediatric age groups, the prevalence of calcification in the adult age group has been evaluated together with the basilar artery, and it has been reported in a wide range from 2% to 13% in different study series [16].

In the Rotterdam Study, the most comprehensive study of the adult population, the prevalence and volume of cervical and intracranial arterial calcification increased with age in both sexes. In that study, age, smoking history, hypercholesterolemia, atherosclerotic vascular diseases, DM, and hypertension were the most important risk factors for calcification. Worldwide, Asian populations have the highest reported prevalence of intracranial arterial calcification [3,17,18].

ICA calcifications evolve with age from a few punctate calcifications or fine linear calcifications to large calcified plaques. It has been shown histologically that calcifications observed in the supraclinoid segment and ICA siphon in the pediatric age group are not associated with atherosclerosis and are located in the internal elastic lamina (IEL) [14,15,16]. Age-related increases in the volume and prevalence of ICA siphon calcification are substantial. Risk factors include excessive alcohol consumption and smoking in men, diabetes mellitus, hypercholesterolemia, history of cardiovascular disease, and early-onset hypertension in women [1,2,3,4,5,6].

Calcifications in intracranial arteries are usually localized in the intima and media of the arterial walls. Focal calcification in damaged or neoplastic brain tissue accounts for at least fifty per cent of all cases of intracranial calcification in all age groups [19]. Dystrophic calcification, particularly in the cerebral cortex, occurs after brain damage due to bacterial meningitis, encephalitis, hypoxic–ischemic injury and, occasionally, ischemic stroke. Vascular anomalies, including arteriovenous malformations and, less commonly, cavernomas, are often focally calcified. In pediatric age groups, cerebral infections caused by Mycobacterium tuberculosis, neurocysticercosis, Cryptococcus neoformans, and, more recently, the Zika virus can lead to many calcified lesions [19].

Calcification of the carotid and vertebral arteries can be seen on cranial and temporal bone CT scans in pediatric age groups. Studies investigating the incidence and pattern of carotid calcification in children are extremely rare in the medical literature. However, the studies that are currently available lack information on vertebral arteries and are out of date [14,19,20,21,22,23].

The aim of this study was to determine the frequency, location, and pattern of incidental carotid and vertebral artery calcifications on cranial CT and temporal bone CT images in children under 15 years of age. We also aimed to investigate the association of these findings with conditions that can cause intracranial calcification and additional risk factors for early atherosclerotic disease, such as hypercholesterolemia, diabetes, and chronic kidney disease.

## 2. Materials and Methods

CT images of the cranial and temporal bone obtained over a 15-month period were retrospectively analyzed to detect calcifications in the carotid and vertebral arteries in patients under 15 years of age. Evaluations included the extent of calcifications within the arteries, their anatomical location, and whether unilateral or bilateral calcifications were present.

After the ICA is separated from the common carotid artery, it is separated into 7 segments as C1–C7 (Figure 1). These are named cervical, petrous, lacerum, cavernous, clinoidal, ophthalmic, and communicating, respectively. C6 and C7 segments can also be classified as supraclinoid segments. Vertebral arteries, which mostly originate from the subclavian artery, are separated into 4 segments as V1–V4 (Figure 2). The first three segments (V1–V3) are extracranial, and the V4 segment is intradural. The V1 segment originates from the subclavian artery and extends to the entrance of the lowest transverse foramen, usually located at the 6th cervical vertebra. The V2 segment crosses the transverse foramen of the 6th cervical vertebra and terminates at the entrance of the transverse foramen of the 1st cervical vertebra. From here, the V3 segment continues until it reaches the dura. This is the beginning of the V4 segment, which continues to form the basilar artery at the pontomedullary junction.

In this study, cranial CT and temporal bone CT images of pediatric age groups in the computed tomography unit of our clinic in 2023 and 2024 were retrospectively reviewed. The anatomical segments of the carotid and vertebral arteries in the non-contrasted cranial CT and temporal bone CT images in consecutive axial–coronal–sagittal planes were evaluated for calcification findings. Twelve patients were excluded from the study due to inadequate evaluation of the bilateral supraclinoid segment and the cavernous and carotid siphon regions of the ICAs (X-ray hardening artefacts, motion artefacts, etc.) and inadequate anatomical or positional imaging of both the vertebral and ICA artery regions. As a result, CT examinations of a total of 300 pediatric patients (156 female, 144 male) were included in the final analysis. Finally, 300 cranial and temporal bone CT scans from 300 pediatric patients were retrospectively evaluated for the presence of carotid and vertebral artery calcification.

Computed tomography acquisition protocols were obtained using a 64-slice dual-source multi-detector CT scanner (Somatom Definition Flash, Siemens Healthcare, Forchheim, Germany) with a slice thickness of 0.1 or 0.2 mm. Image analysis was performed by two radiologists with 11 and 5 years of experience in pediatric imaging. Slices above and below the level of calcification were carefully reviewed to ensure that findings were not due to partial volume averaging of adjacent structures.

The presence of calcifications in the internal carotid and vertebral arteries was graded using a four-point grading scale developed specifically and uniquely for this study (Table 1). On this scale, hyperdense foci less than 1 mm in diameter were considered suspicious calcifications. Hyperdensities greater than or equal to 1 mm in diameter were considered prominent calcifications.

We evaluated the patient group according to this grading scale and included calcifications confirmed by both radiologists in the final analysis. We retrospectively reviewed the medical records of all patients who were found to have definite calcifications in the internal carotid artery and the vertebral artery segments by reviewing patient files. (In our study, we included only patient medical records that are currently available in the database system of our hospital. We did not have access to the medical records of some patients, and we also did not have access to examinations performed at an external center). To investigate possible associations between different pathological conditions associated with calcifications in pediatric patients and these arterial calcifications, we noted pathologies that may predispose to arterial calcifications, such as hypercholesterolemia, chronic kidney disease, and diabetes mellitus, from the patients’ medical data. We also investigated possible associations between intracranial calcifications and various pathological conditions and diseases, listed in Table 2, in patients with ICA and vertebral artery calcifications [19,24]. This included previous cross-sectional radiological examinations of the patients.

### Statistical Analysis

Our study is descriptive. Therefore, descriptive statistics in the form of means, medians and percentiles were used. Advanced statistical methods were not used. Mean ± standard deviation was used as the descriptive statistic for numerical variables and the number and percentage (%) for categorical variables. SPSS 25.0 (IBM Corporation, Armonk, New York, NY, USA) was used for statistical analysis.

In this study, radiologist 1 and radiologist 2, experienced in cross-sectional imaging in pediatric age groups, independently evaluated CT images. Cohen’s kappa coefficient was used to determine whether radiologists 1 and 2 agreed on the presence of pathology and quality assessment. The level of agreement was categorized as follows: *p* < 0.05 was considered statistically significant; a coefficient between 0 and 0.20 was considered low; a coefficient between 0.21 and 0.40 was considered moderate; a coefficient between 0.41 and 0.60 was considered moderate; a coefficient between 0.61 and 0.80 was considered substantial; and a coefficient between 0.81 and 1.00 was considered to be highly compatible.

The Ethics Committee approved our study. The following is the Ethics Committee approval number. Date: 12 December 2023. Number: 2023.12/003-128.6.

All methods used in studies involving human subjects complied with the Helsinki Declaration of 1964 and its subsequent amendments, the ethical standards of the institutional and/or national research committee, or similar ethical standards.

## 3. Results

In this study, cranial and temporal bone CT images of a total of 300 patients were analyzed in detail. The age of the patients ranged from 4 months to 15 years, with a mean age of 8.9 ± 0.2 years. The male to female ratio was approximately 1:1 (156 females, 144 males). Definite ICA and vertebral artery calcifications were seen in 23.6% (71/300) of the patients. Suspected calcifications were seen in 9.3% (28/300) of the analyzed CT sections. The proportion of patients with no evidence of arterial calcification was 67% (201/300). Definite calcifications were seen in 8.4% (6/71) of patients younger than 2 years, 26.7% (19/71) of patients aged 2–7 years, 39.4% (28/71) of patients aged 7–12 years, and 25.3% (18/71) of patients aged 12–15 years. Of the 71 patients with definite calcifications, 43% (31) were male and 56% (40) were female. Of the calcifications, 66% (47) were unilateral and 33% (24) were bilateral. In this study, a total of 300 CT images were analyzed, and definite and suspected foci of calcification were detected in the carotid artery in 82 patients (27.3%) and in the vertebral artery in 17 patients (5.6%). In our study, calcifications in the supraclinoid segment and the carotid siphon regions were observed in 62 patients (20.6%), and these were the most common anatomical locations of ICA calcifications (Table 3). V4 segment calcifications, which were observed in 15 cases (5%), were the most common anatomical location of vertebral artery calcifications (Table 4) (Figure 3, Figure 4, Figure 5 and Figure 6). The most common vascular calcification pattern was focal–punctate calcification (65 patients, 21.6%).

In our study, the agreement between the evaluations of radiologist 1 and radiologist 2 was measured between 0.81 and 1.00 in all cases and found to be highly compatible. For this reason, no image in our study was excluded from the study.

Medical records, clinical follow-up charts, and laboratory results of 71 patients with definite calcifications were available for review in the hospital database system. All medical data and results of these patients were carefully analyzed and recorded. Only two patients were found to have pre-existing and definitively diagnosed type 1 diabetes mellitus, which is known to contribute to early atherosclerosis. Two patients had a history of stage-I germinal matrix and intraventricular hemorrhage according to the Volpe classification in the neonatal period without any other known cause. However, there was no evidence of necrotic damage to the cerebral white matter or brain regions, such as the pons or hippocampus, or cerebral infarction in either patient. An analysis of the clinical findings and laboratory results of 71 patients with definite calcifications in the ICA and vertebral arteries found no evidence of diseases that may predispose to arterial calcifications, such as hypercholesterolemia, chronic kidney disease, and diabetes mellitus. Incident carotid and vertebral artery calcification did not correlate with other diseases.

## 4. Discussion

In retrospective analyses performed to measure the frequency of identifiable calcifications in the carotid artery and vertebral artery walls of pediatric patients who underwent cranial CT and temporal bone CT imaging, it was noted that there was a significant increase of approximately 65% after the age of seven [14]. In a large series of adult patients, Ptak et al. concluded that there was a significant correlation between diabetes mellitus, hypercholesterolemia, and hypertension and the occurrence of carotid artery calcifications [25].

In a study of a patient population over 70 years of age, Katada et al. reported the incidence of incidental carotid and vertebral artery calcifications on cranial CT scans to be 15% [26]. This study also showed a significant association between vertebral artery calcification and aging. Vertebral artery calcification was not found in patients under 40 years of age. It has been reported that vertebral artery calcification started to be seen in patients over 50 years of age, and then the incidence increased with age [26].

B. Koch et al. hypothesized that well-defined foci of hyperdensity in the arterial wall in the ICA supraclinoid segment and the carotid siphon regions represent calcifications, and we accepted these imaging findings as suspicious foci of calcification in our pediatric age group series. And, we concluded that marked hyperdensities in the walls of the carotid arteries were likely to represent calcifications [14].

B. Koch et al. reported that definite calcifications were found in 25% and suspicious calcifications in 27.6% of the ICA structures of 663 patients in the pediatric age group [14]. It has been reported that the incidence of carotid calcifications increases with age in the pediatric age group. And, it has been reported that calcifications are more often localized, especially in the distal segment of the carotid siphon. This is because in the early stages of life, there is not much connective tissue around the carotid siphon, but, by the age of 5 years, this segment of the ICA becomes more rigid due to the increase in connective tissue surrounding the vascular structure, and the arterial vascular structure becomes more adherent to the bone cavity. For incidental vertebral artery calcifications, there is no study in the literature for specific pediatric age groups. Our research study is a first for vertebral arteries in this respect. The prevalence of vertebral artery calcifications in the adult period has been evaluated together with the basilar artery, and it has been reported in a wide range from 2% to 13% in various study series.

EM Peters et al. reported in a study that it is not always possible to exclude the presence of iron deposition in histopathological examinations of different intracranial arterial beds and calcification foci. They suggested the presence of calcifications in hemochromatosis and thalassemia as an example [27]. However, iron deposition usually exceeds 130 Hounsfield units density and can be clearly distinguished from calcifications on CT. We believe that the pathophysiology of calcifications in different vascular beds of the ICA and vertebral artery segments, where calcifications are common, should be the subject of a separate investigation.

John H. Livingston et al. report that intracranial calcifications are a common finding on neuroimaging in pediatric neurology practice and that calcifications occur in damaged, neoplastic, or malformed cerebral structures in approximately half of all cases. However, they do not propose a common physiopathological mechanism for pediatric vascular calcifications. They have not presented a systematic approach for the identification of characteristic vascular bed calcification patterns and their radiologic findings, especially in congenital infections and some genetic diseases. According to their research, congenital CMV infection accounts for a significant percentage of all cases. However, radiological and clinical aspects should be thoroughly assessed before concluding that cerebral calcifications are due to congenital infection, as several genetic diseases can mimic congenital infection, they noted. A methodical approach to identifying radiographic abnormalities allows for a diagnosis to be made in many cases, and characteristic patterns of calcification are observed in many cases. In the same study, John H. Livingston et al. said that some hereditary diseases, the origin of which has not yet been identified, are thought to be associated with cerebral calcification [19].

In our opinion, the main differences between physiological and pathological cases in pediatric ICA and vertebral artery calcifications should be investigated in large series in terms of physiopathological and radiological patterns. We also think that further investigations of the anatomical location and pattern correlations of histologically intracranial vascular bed calcification features in different neuroimaging modalities is an important issue. A comprehensive multidisciplinary approach is required to plan studies that emphasize the importance of whether there are interval changes in vascular calcifications with advancing age and in which cases they show progression.

The “Clinical Practice Guidelines on the Management of Atherosclerotic Carotid and Vertebral Artery Disease” established by the European Society for Vascular Surgery (ESVS) in 2023 aimed primarily to reduce the clinical impact of ICA and vertebral stenoses to prevent transient ischemic attack (TIA) or ischemic stroke [28]. However, there are no studies investigating the presence of plaque structures with calcifications that may lead to carotid and vertebral artery stenosis and their clinical effects in pediatric age groups. The most important factor in this regard is considered to be the infrequency of transient ischemic attacks in pediatric age groups.

Bergevin et al. stated that non-severe calcification may be due to vascular maturation and aging [20]. According to some studies, carotid artery calcification is a well-known sign of atherosclerosis and associated with significant morbidity and mortality. Intimal and medial calcification are the two categories of carotid calcification. It is now known that vascular calcification is an active, enzymatically controlled process involving endothelial dysfunction and early-stage dystrophic calcification. A pathogenic inflammatory response is triggered, leading to the deposition of calcium phosphate in the form of microcalcifications. This, in turn, leads to plaque formation and, ultimately, destabilization with the problems that follow. As the inflammation subsides, macrocalcifications and hydroxyapatite crystal formation take over, keeping the plaque stable. Early detection of carotid artery calcification through imaging has been found to be crucial [21,22,23]

In our study, the lack of association between incidental calcifications detected on CT imaging and risk factors for atherosclerotic disease in children under 15 years of age suggests that these incidental calcifications may be within physiological limits. We examined our study population for diseases associated with intracranial calcifications but found no association with these diseases. Therefore, it was suggested that the incidental calcifications in our study were not indicative of pathology and may develop under physiological conditions, such as arterial pressure differences or vascular maturation. We believe that the significance of incidental carotid artery wall calcifications in children can be analyzed in prospective studies of pediatric patients.

The main limitations of our study are that it was a single-center retrospective study and that the number of patients was limited. The number of patients under the age of 2 years in the database should be increased. In addition, because arterial vessel diameters show significant differences between age groups, we could not perform standardized measurements in this regard in our study, and we consider this to be one of the limitations of our study. In addition, the effect of epidemiological factors, such as ethnicity, on the occurrence of carotid and vertebral artery calcification was not evaluated in this study. Future studies in larger pediatric populations may provide additional information on epidemiological factors.

In addition, efforts are underway to integrate artificial intelligence into CT imaging and to achieve more advanced CT technologies over time. As a result of these studies, incidental carotid artery and vertebral artery calcifications will be easily detected in much larger patient groups with artificial intelligence. This will allow data to be collected in larger populations with lower error rates. We are excited about the new imaging techniques and trends that are constantly evolving in radiology [29].

## 5. Conclusions

Incidental calcifications in the ICA and the vertebral arteries are a common finding in cranial CT and temporal bone CT examinations in the pediatric age group, but they have not been associated with any disease or pathological manifestations. It is thought that these calcifications are not attributable to underlying conditions that may lead to early atherosclerotic calcification but are most likely a physiological response to turbulent flow in the natural anatomical curves of the arterial vascular structures.

As a result of this study, we suggest that calcifications in the supraclinoid segment and siphon of the ICA and calcification foci in the vertebral arteries may begin to develop at an early age, and the degree of ICA calcification may evolve with age from only a few focal punctate or fine linear patterns to large calcific plaques.

In conclusion, this study statistically analyzes the incidence, anatomical location, patterns, and possible etiological factors of incidental carotid and vertebral artery calcifications in pediatric age groups and provides concrete data to consider. These results may help to revive interest in incidental vascular calcifications in pediatric age groups.

## Figures and Tables

**Figure 1 diagnostics-15-01263-f001:**
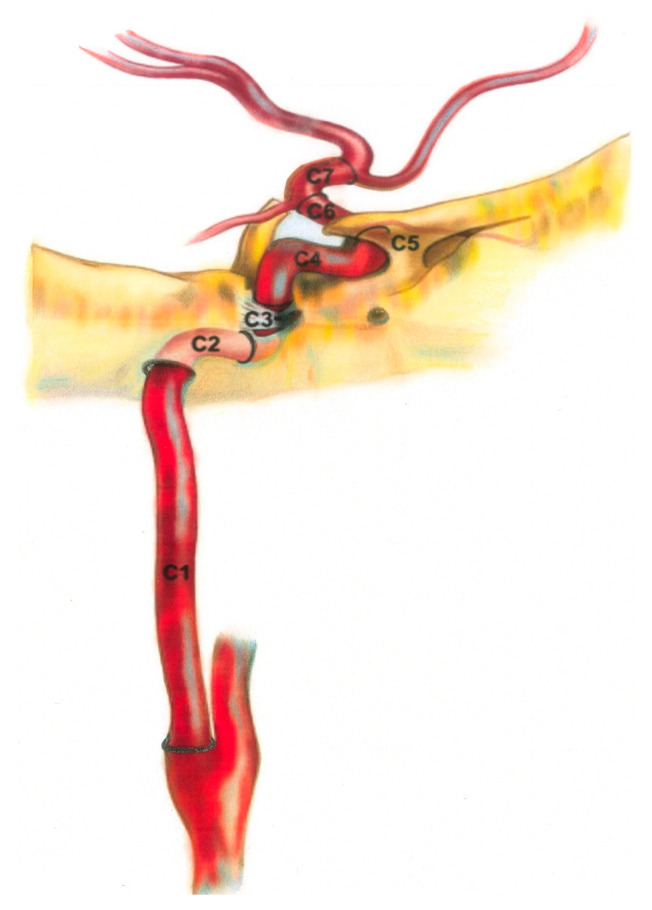
ICA segments.

**Figure 2 diagnostics-15-01263-f002:**
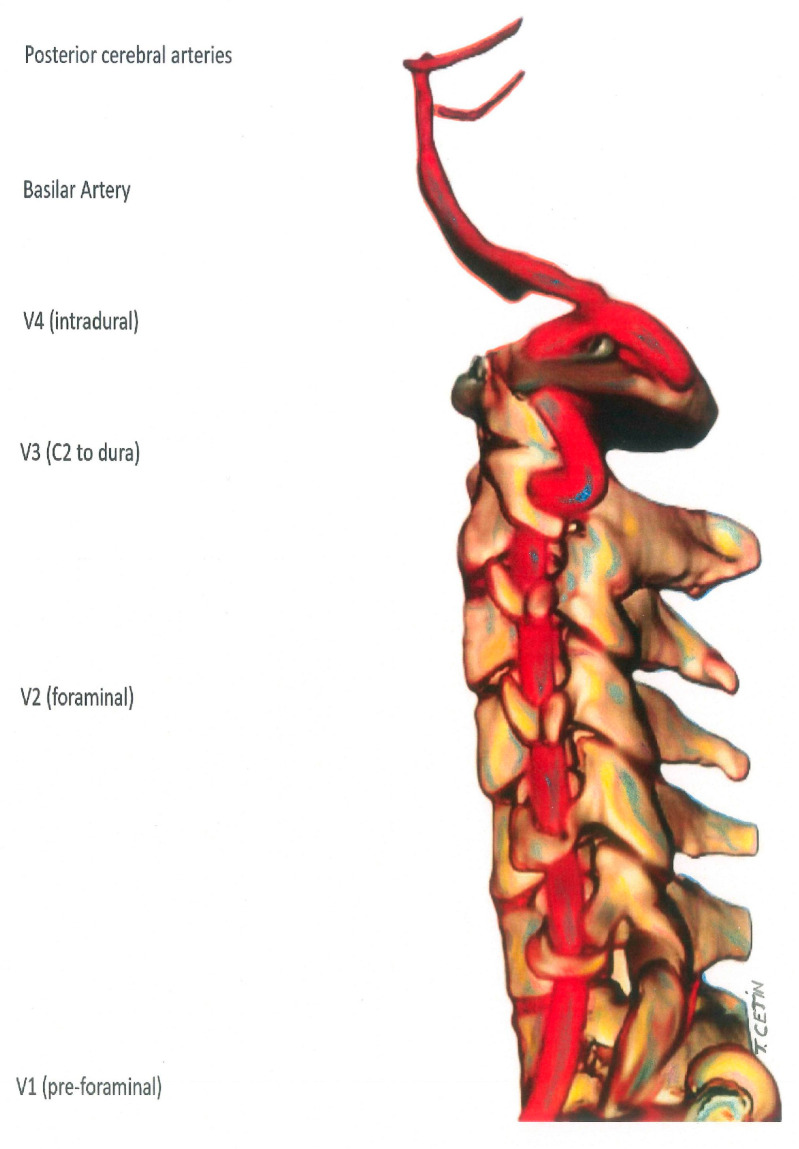
Vertebral artery segments.

**Figure 3 diagnostics-15-01263-f003:**
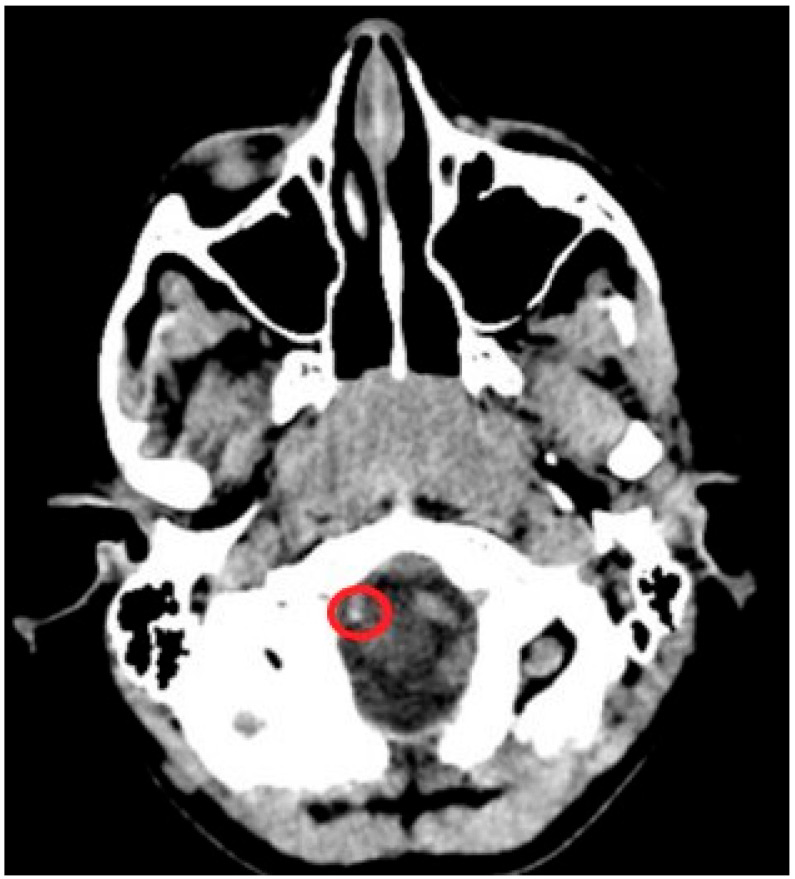
A 5-year-old girl with chronic otitis media; CT imaging of the temporal bone was performed. Hyperdense punctate calcification is shown in the V4 segment of the right vertebral artery (circle).

**Figure 4 diagnostics-15-01263-f004:**
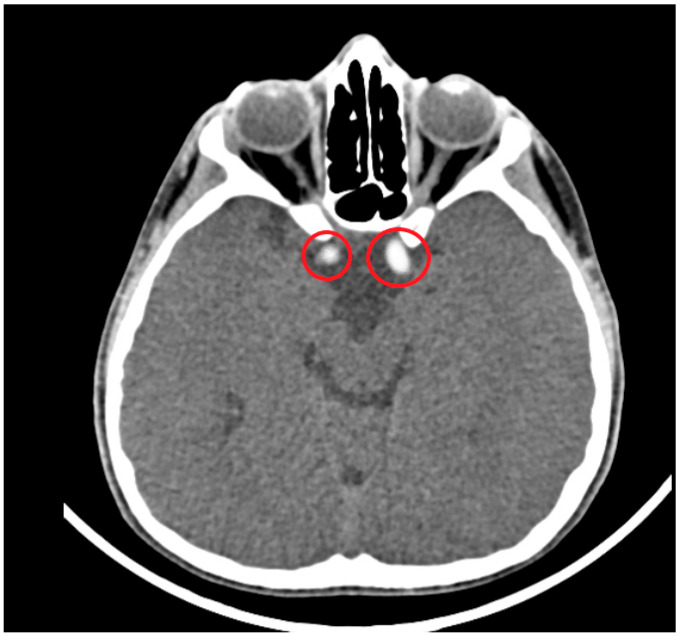
A 7-year-old boy patient had head trauma. Cranial CT shows hyperdense focal punctate calcifications in the supraclinoid segments of bilateral ICAs (circles).

**Figure 5 diagnostics-15-01263-f005:**
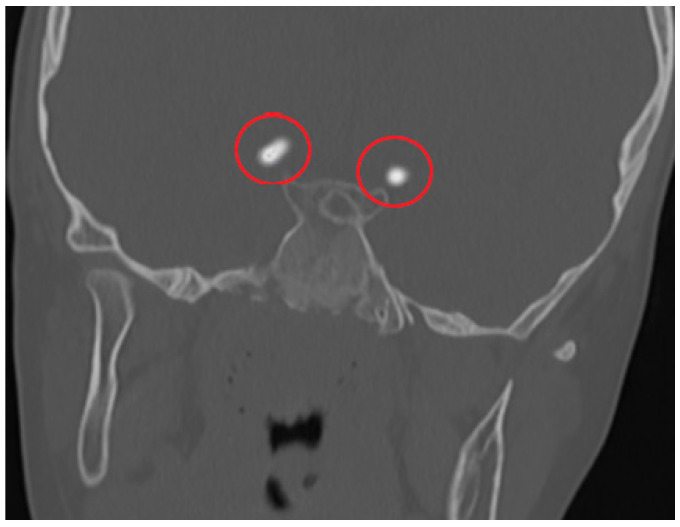
A 9-year-old male patient complained of severe headache and nausea and vomiting. Cranial CT imaging was performed. Hyperdense punctate calcification is shown in the V4 segment of the bilateral vertebral artery (circles).

**Figure 6 diagnostics-15-01263-f006:**
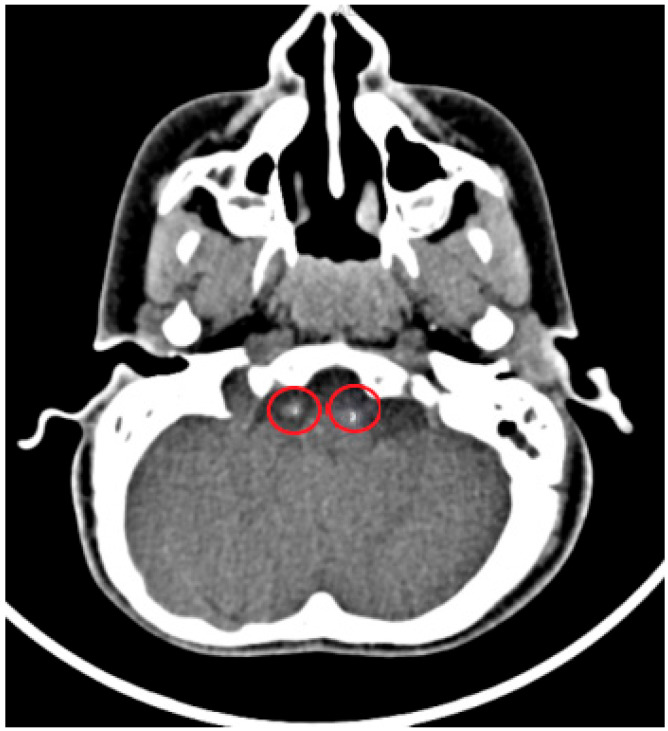
A 6-year-old male patient complained of headache and vomiting. Cranial CT imaging was performed. Hyperdense punctate calcifications are shown in the V4 segment of the bilateral vertebral artery (circles).

**Table 1 diagnostics-15-01263-t001:** Scale for grading the presence of ICA and vertebral artery calcifications (original EBYU Pediatric Classification Scale).

		Number and Percentage of Patients
Category 0	No evidence of calcification	201 (67%)
Category 1	Unilateral suspicious calcification focus	28 (9.3%)
Category 2	Prominent calcifications on one or more images, ranging from a single hyperdense focal point to peripheral areas of calcification	47 (15.6%)
Category 3	Prominent foci of calcification in bilateral ICA or vertebral arteries	24 (8.1%)

**Table 2 diagnostics-15-01263-t002:** Summary of intracranial calcifications in childhood [19,24].

Intracranial Calcification	Etiologies	Anatomic Location/Calcification Pattern
Physiologic/Age-Related Intracranial Calcifications		Pineal gland, choroid plexus, falx cerebri, tentorium cerebelli, basal ganglia
Genetic Syndromes/Developmental Disorders	Sturge–Weber syndrome	Gyriform design with twin lines; similar to a tram track
	Tuberous sclerosis	Tubers that are subcortical and subependymal along the atrium and caudothalamic groove
	Neurofibromatosis	Choroid plexus calcifications in the latteral ventricles and cerebellar nodular calcifications
	Cockayne syndrome	Bilateral prominent or punctate calcifications at the level of the basal ganglia
	Krabbe disease	Corona radiata and internal capsule
	Pseudo-TORCH syndromes	Cortical bands, as well as in the thalamus, pons, and cerebellum
	Hyperphenylalaninemia	Basal ganglia
	Von Hippel–Lindau syndrome	Endolymphatic sac tumor
	Mitochondrial disorders	Dispersed or punctate in the thalamus and basal ganglia
	Fahr disease	Caudate, putamen, globus pallidus, thalamus, deep cortex, and dentate are all symmetrically involved
Congenital Infection	Cytomegalovirus	The basal ganglia have mild punctate calcifications, and the periventricular region has thick, chunky calcifications
	Herpes	Dispersed
	Toxoplasmosis	Nodular calcification is seen in the periventricular and cortical regions Curvilinear calcification is seen in the thalamus and basal ganglia
	Rubella	The periventricular region and basal ganglia
	Zika	Subcortical punctate calcifications
	Human Immunodeficiency Virus	Subcortical tissue and basal ganglia
Acquired Infection	Neurocysticercosis	Calcific nodule within a calcified cyst
	Mycobacterium tuberculosis	Central calcific tuberculomas
	Cryptococcus neoformans	Parenchymal and leptomeningeal punctate calcifications
Vascular Malformations	Arteriovenous malformation	Punctate or curvilinear calcifications may be present
	Cavernous malformation	Amorphous/punctate calcifications
	Developmental venous anomaly	Dystrophic calcifications
Intra-Axial Neoplastic	Pilocytic Astrocytoma	Extensive calcification rarely occurs
	Oligodendroglioma	Nodular and grouped
	Ganglioglioma	Calcific mural nodules
	Medulloblastoma	Dispersed foci or grouped
Extra-Axial Neoplastic	Meningioma	Spherical and rim
	Craniopharyngioma	Thin and peripheral
	Germ cell tumors	Heterogeneous
	Lipoma	Eggshell calcifications
Intraventricular	Ependymoma	Point or mass-like
	Central neurocytoma	Variable, ranging from small punctate foci to large calcifications
Metabolic/Endocrine	Hypoparathyroidism	Basal ganglia
Inflammatory	Systemic lupus erythematosus	Most common in the cerebellum
	Sarcoidosis	Cerebellum, hypothalamus, and suprasellar regions

**Table 3 diagnostics-15-01263-t003:** Sites of occurrence of ICA calcifications.

	Cervical	Petrous	Lacerum	Carotid Siphon	Supraclinoid
ICA Calcifications	7 (2.3%)	7 (2.3%)	6 (2%)	31 (10.3%)	31 (10.3%)

**Table 4 diagnostics-15-01263-t004:** Sites of occurrence of vertebral artery calcifications.

	V1	V2	V3	V4
Vertebral Artery Calcifications	1 (0.33%)	1 (0.33%)	0	15 (5%)

## Data Availability

The data that support the findings of this study are available from the corresponding author (BI) upon reasonable request.

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
