# Peer review of "Incidental Calcifications of Carotid and Vertebral Arteries: Frequency and Associations in Pediatric Population"

_diagnostics, 2025, doi:10.3390/diagnostics15101263_

Round 1
Reviewer 1 Report
Comments and Suggestions for Authors
The manuscript entitled “Incidental Calcifications Of Carotid And Vertebral Arteries: Frequency And Associations In Pediatric Population” showed that the frequency, localization and patterns of incidental carotid and vertebral artery calcifications on cranial CT and temporal bone CT images in children under 15 years of age. Furthermore, this study elucidated the potential associations between these calcifications and various diseases. However, the manuscript needs major revisions before it can be accepted in this journal, as explained below:
Question 1 :Why did the author choose the population aged 15 and below as the research object?
Question 2: Whether gender differences in this study had any effect on the frequency, localization and patterns of incidental carotid and vertebral artery calcifications on cranial CT and temporal bone CT images?If the answer is yes, how did the author avoid these effects?
Question 3 :Do changes in calcification with age lead to any disease or pathological manifestations in paediatric age group?
Question 4 :Given the manuscript's reliance on cranial and temporal bone CT imaging , how might findings differ in other techniques include MRI or ultrasonic imaging?
Question 5 :It is recommended to check whether the title of the form should be placed above the form.
Comments on the Quality of English LanguageThe English writing is acceptable.
Author Response
Dear Editor,
Thank you for giving us the opportunity to submit a revised draft of the manuscript. We appreciate the time and effort that you and the reviewers dedicated to providing feedback on our manuscript and are grateful for the insightful comments on and valuable improvements to our paper. We have incorporated most of the suggestions made by the reviewers. Please see below for a point-by-point response to the reviewers’ comments and concerns.
Reviewer 1 report:
Comments 1:
Why did the author choose the population aged 15 and below as the research object?
Response 1:
We wanted to perform our study in patients under the age of 18. However, we found that the number of patients in the 16-18 age group was too low to be statistically evaluated when patients with deterioration in image quality due to artefacts and patients in whom adequate anatomical imaging could not be performed were excluded from the study. For this reason, we limited our study group to patients under 15 years of age.
Comments 2:
Whether gender differences in this study had any effect on the frequency, localization and patterns of incidental carotid and vertebral artery calcifications on cranial CT and temporal bone CT images?If the answer is yes, how did the author avoid these effects?
Response 2:
Gender differences had no effect on the frequency, localisation and patterns of incidental carotid and vertebral artery calcifications.
Comments 3:
Do changes in calcification with age lead to any disease or pathological manifestations in paediatric age group?
Response 3:
In our study, no association was found between incidental calcifications and any disease or pathological change. However, pathological conditions seen together with intracranial calcifications are listed in Table 2.
Comment 4:
Given the manuscript's reliance on cranial and temporal bone CT imaging , how might findings differ in other techniques include MRI or ultrasonic imaging?
Response 4:
All segments of the vertebral artery and carotid artery can be evaluated on CT examination. However, only the extracranial proximal segments of the vertebral artery and carotid artery can be analysed by ultrasonography. Therefore, a statistically sound comparison cannot be made with CT examination. MR examination is not as successful as CT examination in showing calcifications. Therefore, CT examination is the most successful test in the examination of incidental calcifications.
Comment 5:
It is recommended to check whether the title of the form should be placed above the form.
Response 5:
We apologise, we could not understand what you meant by this comment.

Reviewer 2 Report
Comments and Suggestions for Authors Very interesting article A specialized topic but of growing importance A slightly radiological / internal medicine vision of the topic You mention that there are markers of cardiovascular calcifications such as lipoprotein A Calcifications such as coronary and aortic calcifications are of growing interest in adult cardiovascular risk stratification A little more attention to the figures, black arrow heads on a black and white CT scan are confusing, isn't a circle like in figure 5 better? I would invest more on the iconographic part by adding some other examples in figures 3 and 4 (perhaps other images adding more cases). Cite this article on the usefulness of the calcium score (coronary) also in primary prevention DOI: 10.1016/j.jjcc.2024.12.005Author Response
Dear Editor,
Thank you for giving us the opportunity to submit a revised draft of the manuscript. We appreciate the time and effort that you and the reviewers dedicated to providing feedback on our manuscript and are grateful for the insightful comments on and valuable improvements to our paper. We have incorporated most of the suggestions made by the reviewers. Please see below for a point-by-point response to the reviewers’ comments and concerns.
Comment 1:
Very interesting article A specialized topic but of growing importance A slightly radiological / internal medicine vision of the topic You mention that there are markers of cardiovascular calcifications such as lipoprotein A Calcifications such as coronary and aortic calcifications are of growing interest in adult cardiovascular risk stratification A little more attention to the figures, black arrow heads on a black and white CT scan are confusing, isn't a circle like in figure 5 better? I would invest more on the iconographic part by adding some other examples in figures 3 and 4 (perhaps other images adding more cases). Cite this article on the usefulness of the calcium score (coronary) also in primary prevention DOI: 10.1016/j.jjcc.2024.12.005
Response 1:
We replaced the arrowheads in Figure 4 with circles. We have added figure 6 as a new figure. We cited the article ‘DOI: 10.1016/j.jjcc.2024.12.005’ with reference number 6.
Reviewer 3 Report
Comments and Suggestions for Authors
The manuscript is devoted to an interesting topic. However, each section of the manuscript requires significant revision.
Abstract
In contrast to the Materials and Methods section, it is unclear if 300 CT images correspond to 300 patients.
Abstract Conclusion section mentions the lack of correlation between carotid or vertebral artery calcifications and concomitant diseases. However, this is not reflected in the Results section.
Introduction
This section is very lengthy and should be shortened to make it more focused.
Materials and Methods
It is stated that images were analyzed by 2 blinded radiologists. However, this information is meaningless because the authors do not report differences or similarities in their assessment (level of agreement) anywhere.
Statistical Methods
Authors describe different statistical methods. However, they do not present any results obtained by these methods besides the percentages. Definition "almost perfect" is unlikely appropriate for characterization of a correlation coefficient.
Results
It is unlikely appropriate to present the list of diseases/conditions checked for association with calcification in the Results section. It is more logical to have it in Materials and Methods.
It is unclear if there were patients with calcification of combined calcification of the vertebral and carotid artery.
Authors state that there was no relationship between intracranial calcification and concomitant diseases. However, no statistical evidence is presented to confirm this statement.
The major limitation that significantly limits the scientific value of the manuscript is the absence of any quantitative data that would characterize the condition of vertebral/carotid arteries.
Discussion
The section is lengthy and not focused.
The statement that discovered calcification "may be within physiological limits requires more detailed explanation.
Conclusion
This section should be concise and do not repeat the Discussion. It is more appropriate to describe limitation of the study in the Discussion section.
Comments on the Quality of English Language
The quality of English language is insufficient. The manuscript should be revised by a translator or a native speaker.
Author Response
Dear Editor,
Thank you for giving us the opportunity to submit a revised draft of the manuscript. We appreciate the time and effort that you and the reviewers dedicated to providing feedback on our manuscript and are grateful for the insightful comments on and valuable improvements to our paper. We have incorporated most of the suggestions made by the reviewers. Those changes are highlighted with track changes function through the manuscript. Please see below for a point-by-point response to the reviewers’ comments and concerns.
Comment 1:
The manuscript is devoted to an interesting topic. However, each section of the manuscript requires significant revision.
Abstract
In contrast to the Materials and Methods section, it is unclear if 300 CT images correspond to 300 patients.
Abstract Conclusion section mentions the lack of correlation between carotid or vertebral artery calcifications and concomitant diseases. However, this is not reflected in the Results section.
Introduction
This section is very lengthy and should be shortened to make it more focused.
Materials and Methods
It is stated that images were analyzed by 2 blinded radiologists. However, this information is meaningless because the authors do not report differences or similarities in their assessment (level of agreement) anywhere.
Statistical Methods
Authors describe different statistical methods. However, they do not present any results obtained by these methods besides the percentages. Definition "almost perfect" is unlikely appropriate for characterization of a correlation coefficient.
Results
It is unlikely appropriate to present the list of diseases/conditions checked for association with calcification in the Results section. It is more logical to have it in Materials and Methods.
It is unclear if there were patients with calcification of combined calcification of the vertebral and carotid artery.
Authors state that there was no relationship between intracranial calcification and concomitant diseases. However, no statistical evidence is presented to confirm this statement.
The major limitation that significantly limits the scientific value of the manuscript is the absence of any quantitative data that would characterize the condition of vertebral/carotid arteries.
Discussion
The section is lengthy and not focused.
The statement that discovered calcification "may be within physiological limits requires more detailed explanation.
Conclusion
This section should be concise and do not repeat the Discussion. It is more appropriate to describe limitation of the study in the Discussion section.
Response:
In the materials and methods section of the abstract, we added the following statement: “300 cranial and temporal bone CT imaging of 300 pediatric patients were retropectively evaluated to detect the presence of calcification in the carotid and vertebral arteries.” We used the same expression in the materials and methods section of the article.
In the results section of the abstract, we added the following statement: “Incidental calcifications of the carotid and vertebral arteries did not significantly correlate with other diseases.”
Thank you for your valuable comment and attention. We have removed the phrase “2 blinded radiologists”.
Instead of “almost perfect” we used “almost fully compatible”.
We presented Table 2 in the materials and methods section. We edited the last sentence of the Materials and methods section as “We also possible relationships between intracranial calcifications and different pathological conditions and diseases listed in Table 2, were investigated in patients with ICA and vertebral arterial calcifications, including previous cross-sectional radiological examinations of the patients [20, 25].”
Thank you for your valuable comment. In our study, we created the EBYU calcification scale and graded according to this scale, but we did not share detailed data on whether there was combined calcification in the carotid and vertebral arteries. This did not have any effect on the course of our study because our study aimed to determine how often calcifications in the vertebral artery and carotid artery are seen, in which anatomical localizations they are seen and whether they are associated with other diseases.
In our study, patients with definite evidence of calcification were examined for the diseases listed in table 2 and the diseases listed in table 2 were not detected in our study population. Therefore, it is stated that there was no association between intracranial calcifications and comorbidities in our study group. Since no association was found, no statistical analysis information was given.
In our study, the EBYU calcification scale was created to provide quantitative data to characterize the condition of the vertebral/carotid arteries. 2 radiologists performed separate examinations to increase the quantitative value of the EBYU calcification scale. The measurements of the two radiologists were consistent with each other and quantitative data on the most common locations and frequency of incidental calcifications were obtained.
To make the expression “may be within physiological limits” more descriptive, we added the sentence “Because our study population was examined for diseases associated with intracranial calcifications, but no association was found with these diseases. Therefore, it was thought that the incidental calcifications in our study did not indicate any pathology and may develop within physiologic conditions such as pressure differences in arteries or vascular maturation.”
We explained the limitations of the study in the discussion section. We apologize for this mistake.
We tried to express the conclusion in a more concise, thank you very much.
Reviewer 4 Report
Comments and Suggestions for Authors
I am grateful to the editor for the opportunity to review the manuscript of Turkhun Cetin et al. "INCIDENTAL CALCIFICATIONS OF CAROTID AND VERTEBRAL ARTERIES: FREQUENCY AND ASSOCIATIONS IN PEDIATRIC POPULATION". In this article, the authors analyzed the frequency of detection of calcification of the carotid and vertebral arteries in children. There are few such studies, especially for the vertebral arteries. The authors tried to answer the question - how often is calcification of these arteries detected and what is its detection associated with? They managed to answer the first question, while answering the second, they did not reveal any factors associated with arterial calcification. Perhaps this is a useful fact that will help to better interpret MSCT data in children in the future.
While reviewing the article, I had the following comments and questions:
1. The Introduction section should be corrected. For example, the description of the anatomical division of the carotid and vertebral arteries into segments (lines 61-79) should be removed from this section. They can be placed in section 2. Materials and Methods.
2. The authors of the article write in the Introduction section: "On cranial and temporal bone CT imaging in pediatric age groups, calcifications in the carotid and vertebral arteries can be seen. Studies examining the incidence and pattern of calcifications in children's carotid arteries are extremely rare in the medical literature. However, the studies that are currently accessible lack information on vertebral arteries and are out of date" (lines 113-117). After this text, it would be appropriate to provide references to the studies from which this information was obtained.
3. In general, the Introduction section needs to be shortened.
4. Information about the ethical issues of the study should be removed from section 2.1 Statistical analysis (lines 179-183). They should be specified in other subsections of Section 2. Materials and Methods.
5. The authors presented their data on the frequency of detection of calcifications as Mean ± standard deviation (on lines 191-206 in the text). This format of presentation of categorical variables is incorrect, it is enough to present the absolute number and percentage.
6. I did not understand the meaning of the presentation of Table 2 in the article. The authors did not indicate in what number of children in their study certain causes of calcification of the carotid and vertebral arteries were encountered. The statement "There was not an apparent statistically significant relationship between intracranial calcifications and all pathological conditions associated with ICA and vertebral artery calcifications" (lines 221-223) is not enough in a scientific article. It is necessary to provide data from statistical analysis.
7. The Limitations of the Study section should be placed in the Discussion section.
8. The Conclusion should be shortened, focusing on the results obtained.
9. The discussion section should begin with the main result obtained by the authors.
Author Response
Dear Editor,
Thank you for giving us the opportunity to submit a revised draft of the manuscript. We appreciate the time and effort that you and the reviewers dedicated to providing feedback on our manuscript and are grateful for the insightful comments on and valuable improvements to our paper. We have incorporated most of the suggestions made by the reviewers. Those changes are highlighted with track changes function through the manuscript. Please see below for a point-by-point response to the reviewers’ comments and concerns.
Comment 1:
I am grateful to the editor for the opportunity to review the manuscript of Turkhun Cetin et al. "INCIDENTAL CALCIFICATIONS OF CAROTID AND VERTEBRAL ARTERIES: FREQUENCY AND ASSOCIATIONS IN PEDIATRIC POPULATION". In this article, the authors analyzed the frequency of detection of calcification of the carotid and vertebral arteries in children. There are few such studies, especially for the vertebral arteries. The authors tried to answer the question - how often is calcification of these arteries detected and what is its detection associated with? They managed to answer the first question, while answering the second, they did not reveal any factors associated with arterial calcification. Perhaps this is a useful fact that will help to better interpret MSCT data in children in the future.
While reviewing the article, I had the following comments and questions:
1. The Introduction section should be corrected. For example, the description of the anatomical division of the carotid and vertebral arteries into segments (lines 61-79) should be removed from this section. They can be placed in section 2. Materials and Methods.
2. The authors of the article write in the Introduction section: "On cranial and temporal bone CT imaging in pediatric age groups, calcifications in the carotid and vertebral arteries can be seen. Studies examining the incidence and pattern of calcifications in children's carotid arteries are extremely rare in the medical literature. However, the studies that are currently accessible lack information on vertebral arteries and are out of date" (lines 113-117). After this text, it would be appropriate to provide references to the studies from which this information was obtained.
3. In general, the Introduction section needs to be shortened.
4. Information about the ethical issues of the study should be removed from section 2.1 Statistical analysis (lines 179-183). They should be specified in other subsections of Section 2. Materials and Methods.
5. The authors presented their data on the frequency of detection of calcifications as Mean ± standard deviation (on lines 191-206 in the text). This format of presentation of categorical variables is incorrect, it is enough to present the absolute number and percentage.
6. I did not understand the meaning of the presentation of Table 2 in the article. The authors did not indicate in what number of children in their study certain causes of calcification of the carotid and vertebral arteries were encountered. The statement "There was not an apparent statistically significant relationship between intracranial calcifications and all pathological conditions associated with ICA and vertebral artery calcifications" (lines 221-223) is not enough in a scientific article. It is necessary to provide data from statistical analysis.
7. The Limitations of the Study section should be placed in the Discussion section.
8. The Conclusion should be shortened, focusing on the results obtained.
9. The discussion section should begin with the main result obtained by the authors.
Response:
- We have edited the introduction section and moved the relevant sections to the materials and methods section. Thank you
- We added a reference to the relevant section. “On cranial and temporal bone CT imaging in pediatrice age groups, calcifications in the carotid and vertebral arteries can be seen. Studies examining the incidence and pattern of calcifications in children's carotid arteries are extremely rare in the medical literature. However, the studies that are currently accessible lack information on ver-tebral arteries and are out of date [14, 20, 21, 22, 23, 24]”
- We have made some shortenings in the introduction.
- We have explained the ethical issues in the materials and methods section.
- We presented categorical variables as absolute numbers and percentages, thank you.
- In our study, patients with definite evidence of calcification were examined for the diseases listed in table 2 and the diseases listed in table 2 were not detected in our study population. Therefore, it is stated that there was no association between intracranial calcifications and comorbidities in our study group. Since no association was found, no statistical analysis information was given. We have moved Table 2 to the materials and methods section.
- We explained the limitations of the study in the discussion section. We apologize for this mistake.
- We tried to express the conclusion in a more concise, thank you very much.
Round 2
Reviewer 1 Report
Comments and Suggestions for Authors
I do not have other concerns, thank you for your revision, and good luck.
Author Response
Comment 1: I do not have other concerns, thank you for your revision, and good luck.
Response 1: Thank you very much for your valuable comments and for your time.
Reviewer 3 Report
Comments and Suggestions for Authors
The authors tried their best to respond to the comments. However, not all of them were properly addressed. Overall, the revisions did not result in a significant improvement in the quality of the paper. Moreover, the quality of English language has become worse. Particular comments are presented below.
Abstract
Frequency of vertebral artery calcification differs from provided in the text (5.7% vs. 5.6%). I would recommend keep the format used in the Abstract across the manuscript (5.6% and not %5.6)
Materials and methods
The authors did not fully understand my comment regarding radiologists and simply deleted the word "blinded". The comment was different. The authors measured the level of agreement between the radiologists using different coefficients. However, they never present the data on these coefficients in the Results section. They stated that only calcifications confirmed by both radiologists were included to the analysis. Does it mean that in all cases the coefficient was between 0.81 and 1.00 and no images were excluded because a disagreement in evaluation. This should be clarified in the manuscript.
It is unclear what do authors mean by "private medical records" they did not have access to, or by "prospective radiological, hematological or biochemical laboratory studies" when their study was retrospective.
Statistical Methods
The authors did not fully address my comment about statistical methods used but only replaced the definition "almost perfect" by "almost fully compatible" which is not appropriate as well because of using "almost". However, my concern was different. The authors state that Fisher’s exact test or Pearson’s Chi-square test was used to assess correlation. However, they do not present any results received by these tests. Overall, the manuscript is free of any statistical analysis. The authors present only percentages and mean age of patients. The authors simply state that the majority of patients did not have diseases that may predispose to calcification. In this situation aforementioned statistical tests are not needed.
Results
It is unclear why the authors did not respond to my question if there were patients with calcifications in both carotid and vertebral artery.
I am not fully satisfied with the response about quantitative assessment of calcifications. The scale using definition of "suspicious" calcification or "prominent" calcification does not provide valuable information if it is not defined what does "suspicious" or "prominent" mean. Moreover, the authors correctly mention that "calcification has a significant negative predictive value for carotid bifurcation stenosis", however, no information about diameters of any arterial segments is not provided.
The authors state that "the male to female ratio was approximately 1:1". It is more appropriate to present the actual number of male and female patients.
The biggest concern is that the manuscript does not present any valuable information except the frequency of calcifications. This information itself does not have a significant clinical importance. The authors mention that calcification may increase with age. However, they did not try to investigate this possible phenomenon. The study with a wide range of ages (from 4-month-old infant to 15-year-old adolescent does provide this opportunity).
Comments on the Quality of English Language
The quality of English language is poor and must be improved. Numerous sentences cannot be clearly understood, or they are written with grammar mistakes. Couple of examples are provided below:
"We also possible relationships between intracranial calcifications and different pathological conditions and diseases listed in Table 2, were investigated in patients with ICA and vertebral arterial calcifications, including previous cross-sectional radiological examinations of the patients".
"Because our study population was examined for diseases associated with intracranial calcifications, but no association was found with these diseases".
Author Response
Comment:
Abstract
Frequency of vertebral artery calcification differs from provided in the text (5.7% vs. 5.6%). I would recommend keep the format used in the Abstract across the manuscript (5.6% and not %5.6)
Materials and methods
The authors did not fully understand my comment regarding radiologists and simply deleted the word "blinded". The comment was different. The authors measured the level of agreement between the radiologists using different coefficients. However, they never present the data on these coefficients in the Results section. They stated that only calcifications confirmed by both radiologists were included to the analysis. Does it mean that in all cases the coefficient was between 0.81 and 1.00 and no images were excluded because a disagreement in evaluation. This should be clarified in the manuscript.
It is unclear what do authors mean by "private medical records" they did not have access to, or by "prospective radiological, hematological or biochemical laboratory studies" when their study was retrospective.
Statistical Methods
The authors did not fully address my comment about statistical methods used but only replaced the definition "almost perfect" by "almost fully compatible" which is not appropriate as well because of using "almost". However, my concern was different. The authors state that Fisher’s exact test or Pearson’s Chi-square test was used to assess correlation. However, they do not present any results received by these tests. Overall, the manuscript is free of any statistical analysis. The authors present only percentages and mean age of patients. The authors simply state that the majority of patients did not have diseases that may predispose to calcification. In this situation aforementioned statistical tests are not needed.
Results
It is unclear why the authors did not respond to my question if there were patients with calcifications in both carotid and vertebral artery.
I am not fully satisfied with the response about quantitative assessment of calcifications. The scale using definition of "suspicious" calcification or "prominent" calcification does not provide valuable information if it is not defined what does "suspicious" or "prominent" mean. Moreover, the authors correctly mention that "calcification has a significant negative predictive value for carotid bifurcation stenosis", however, no information about diameters of any arterial segments is not provided.
The authors state that "the male to female ratio was approximately 1:1". It is more appropriate to present the actual number of male and female patients.
The biggest concern is that the manuscript does not present any valuable information except the frequency of calcifications. This information itself does not have a significant clinical importance. The authors mention that calcification may increase with age. However, they did not try to investigate this possible phenomenon. The study with a wide range of ages (from 4-month-old infant to 15-year-old adolescent does provide this opportunity).
Comments on the Quality of English Language
The quality of English language is poor and must be improved. Numerous sentences cannot be clearly understood, or they are written with grammar mistakes. Couple of examples are provided below:
"We also possible relationships between intracranial calcifications and different pathological conditions and diseases listed in Table 2, were investigated in patients with ICA and vertebral arterial calcifications, including previous cross-sectional radiological examinations of the patients".
"Because our study population was examined for diseases associated with intracranial calcifications, but no association was found with these diseases".
Response:
Thank you very much for your valuable comments and for your time.
We corrected the error in the frequency of vertebral artery calcification and edited the format of the percentages to be the same throughout the text.
“In the current study, 300 CT images were analyzed, and calcification was found in the vertebral artery in 17 patients (%5.6) and the carotid artery in 82 patients (%27.3). The supraclinoid segment is the most common location of carotid artery calcifications with 62 patients (%20.7).”
We have added “In our study, the agreement between the evaluations of radiologist 1 and radiologist 2 was measured between 0.81 and 1.00 in all cases and was found to be highly compatible. For this reason, no image in our study was excluded from the study.” to the relevant section.
There was a mistake in the expression ‘We were unable to review private medical record and did not have access to prospective radiological, haematological or biochemical laboratory studies on these patient subgroups.” and the situation was understood differently. We apologise for this mistake. We have removed the relevant expression and corrected the preceding sentence as ‘(In our study, we included only the patient medical records that are currently available in the database system of our hospital. We did not have access to the medical records of some patients, and we also did not have access to examinations performed at an external centre.)”
We understood what was wanted to be explained about statistical analysis. Thank you very much. In the relevant section, we removed the sentence “At the qualitative measurement level, the correlation of the features with each other was examined with Fisher’s exact test or Pearson’s Chi-square test.” and added the sentence “Our study is descriptive in nature. For this reason, descriptive statistics in the form of mean, median and percentiles were used. Advanced statistical methods were not used.” instead.
There are patients with calcifications in both carotid and vertebral arteries, but this was not part of our study. In our study, we briefly investigated the anatomical localisations where calcifications are common in both carotid and vertebral arteries. We considered the possibility that these calcifications may be associated with any disease. Therefore, the number of patients with calcifications in both arteries is beyond the scope of our study. We did not include additional statistical information on this subject.
In the materials and methods section, we added explanatory statements for the words “suspicious” and “ prominent” in our calcification assessment scale.
“On this scale, hyperdense foci less than 1 mm in diameter were considered suspicious calcification. Hyperdense foci greater than or equal to 1 mm in diameter were considered prominent calcification.”
We have indicated the number of male and female patients where relevant.
“The male to female ratio was approximately 1:1 (156 female, 144 male).”
Reviewer 4 Report
Comments and Suggestions for Authors
The authors responded to my comments and questions and made corrections to the text of the manuscript. I have no other comments.
Author Response
Comment 1: The authors responded to my comments and questions and made corrections to the text of the manuscript. I have no other comments.
Response 1: Thank you very much for your valuable comments and for your time.
Round 3
Reviewer 3 Report
Comments and Suggestions for Authors
Unfortunately, cosmetic changes in the manuscript does not improve the validity.
Comments on the Quality of English LanguageThe quality of English still requires improvement. Even the revised sentences are incorrect and cannot be clearly understood (e.g., "Because we examined our study population for diseases associated with intracranial calcifications, but we did not find any association with these diseases").
Author Response
Dear editor, thank you very much for your time. We have made arrangements to improve the quality of English and have obtained a language certificate. We are sending it in additional files.
